

# Cloud droplet activation of black carbon particles coated with organic compounds of varying solubility

Maryam Dalirian[1], Arttu Ylisirniö[2], Angela Buchholz[2], Daniel Schlesinger[1], Johan Ström[1], Annele Virtanen[2] and Ilona Riipinen[1]

[1] Department of Environmental Science and Analytical Chemistry (ACES) and the Bolin Centre for Climate research, Stockholm University, Stockholm, Sweden

[2] Department of Applied Physics, University of Eastern Finland, Kuopio, Finland

*Correspondence to:*

Maryam Dalirian (maryam.dalirian@aces.su.se)

Ilona Riipinen (ilona.riipinen@aces.su.se)

**Abstract:**

Atmospheric black carbon (BC) particles are a concern due to their impact on air quality and climate. Their net climate effect is, however, still uncertain. This uncertainty is partly related to the contribution of coated BC-particles to the global CCN budgets. In this study, laboratory measurements were performed to investigate cloud condensation nuclei (CCN) activity of BC (Regal black) particles, in pure state or coated through evaporating and subsequent condensation of glutaric acid, levoglucosan (both water-soluble organics) or oleic acid (an organic compound with low solubility). A combination of Soot Particle Aerosol Mass Spectrometer (SP-AMS) measurements and size distribution measurements with Scanning Mobility Particle Sizer (SMPS) showed that the studied BC particles were nearly spherical agglomerates with a fractal dimension of 2.79 and that they were coated evenly by the organic species. The CCN activity of BC particles increased after coating with all the studied compounds and was governed by the fraction of organic material. The CCN activation of the BC particles coated by glutaric acid and levoglucosan were in good agreement with the theoretical calculations using shell-and-core model, which is based on a combination of the CCN activities of the pure compounds. The oleic acid coating enhanced the CCN activity of the BC particles, even though the pure oleic acid particles were CCN inactive. The surprising effect of oleic acid might be related to the arrangement of the oleic acid molecules on the surface of the BC cores or other surface phenomena facilitating water condensation onto the coated particles. Our results show potential in accurately predicting the CCN activity of atmospheric BC coated with organic species by present theories, given that the identities and amount of the coating species



are known. Furthermore, our results suggest that even relatively thin soluble coatings (around 2 nm for the compounds studied here) are enough to make the insoluble BC particles CCN active at typical atmospheric supersaturations and thus be efficiently taken up by cloud droplets. This highlights the need of an accurate description of the composition of atmospheric particles containing BC to unravel their net impact on climate.

*Keywords*: Black carbon, CCN activation, coated aerosols

## 1   Introduction

The effects of aerosols on climate and public health are central topics in atmospheric and environmental research. Atmospheric aerosols influence the energy balance of the Earth and the climate directly by scattering and absorbing solar and thermal radiation (McCormick and Ludwig, 1976; Haywood and Boucher, 2000; Ramanathan et al., 2001; Myhre et al., 2013) and indirectly by acting as cloud condensation nuclei (CCN) and ice nuclei (IN) and changing cloud microphysical properties (Twomey, 1974; Lohmann and Feichter, 2005; Boucher et al., 2013; Myhre et al., 2013). High concentration airborne

particulate matter can also harm human health, and causes millions of premature deaths every year (Mackay and Mensah, 2004; Pope and Dockery, 2006; Pope et al., 2009; Tranfield and Walker, 2012; Slezakova et al., 2013). Particles containing BC are of special importance because of their contribution to global warming. On one hand, they are estimated to be among the most important anthropogenic positive climate forcers after carbon dioxide and methane (Bond et al., 2013; Myhre et al., 2013) through their light-absorbing properties. At the same time their contribution to CCN budgets is not well understood and

drastically influenced by the mixing state of the atmospheric aerosol (Bond et al., 2013). BC-containing particles are emitted into the atmosphere from natural and anthropogenic sources through incomplete combustion of fossil fuels, biomass, and biofuels (Bond et al., 2004). Freshly emitted BC particles are typically fractal aggregates and usually contain co-emitted organics such as adsorbed polyaromatic hydrocarbons (PAHs) produced during combustion (Marr et al., 2006). After emission, they undergo different aging processes and adsorb other organic and inorganic material produced by gas-to-particle

conversion processes (Zhang et al., 2008; Canagaratna et al., 2015).

        The net climate forcing caused by BC-containing particles is still uncertain, due to e.g. lack of knowledge about cloud interactions of BC-containing particles and the role of the co-emitted species. To overcome these uncertainties, several studies have recently investigated the structure, hygroscopic growth and CCN activation of BC mixed with other, usually more water-soluble, species. Some of these studies have indicated that by increasing the amount of the coating on the BC particles, the

dynamic shape factor of these particles decreases, fractal BC aggregates become restructured and more compact (Saathoff et al., 2003; Slowik et al., 2007; Zhang et al., 2008; Pagels et al., 2009; Tritscher et al., 2011; Ghazi and Olfert, 2012). The





hygroscopic growth factors (HGF) and CCN activation of BC particles coated with various species have also been investigated in other studies (Saathoff et al., 2003; Dusek et al., 2006; Hings et al., 2008; Zhang et al., 2008; Henning et al., 2010, 2012; Stratmann et al., 2010; Tritscher et al., 2011; Maskey et al., 2017). These studies showed that the adsorption of water onto the slightly soluble part of the coated particles increases water uptake and hygroscopic growth factors (HGF) and facilitates the process of adsorption activation (CCN activity). Despite the mentioned studies, CCN activation measurements of BC particles containing a soluble fraction are still relatively scarce.

The CCN activation of uncoated and coated insoluble particles, such as BC-particles coated with soluble species, is usually described theoretically by multilayer adsorption models accounting for the curvature of the particles. One of these theories is adsorption activation theory (Sorjamaa and Laaksonen, 2007; Kumar et al., 2009) which is a combination of FHH (Frenkel, Halsey and Hill) adsorption isotherms and classical Köhler theory to describe the equilibrium growth of insoluble particles. Later, Kumar et al. (2011) introduced a new framework of CCN activation of dust containing a soluble salt fraction, based on a combination of the classical Köhler and FHH adsorption theories. Meanwhile, the development and deployment of Soot Particle Aerosol Mass Spectrometer (SP-AMS) in the recent years has resulted in enhanced knowledge about the composition and structure of fresh and aged BC-containing particles (Onasch et al., 2012; Willis et al., 2014; Canagaratna et al., 2015), but uncertainties still remain about the exact characteristics of ambient BC, related to e.g. morphology of the freshly emitted and aged BC particles and their importance for aerosol-cloud interaction.

The main aim of this study was to gain further insight into the structure and CCN activity of BC particles with various degrees of coating with atmospheric organic molecules. Laboratory measurements were performed to study the CCN activity of the pure and coated BC particles and the experimental CCN activity results were compared to theoretical calculations using the framework introduced by Kumar et al. (2011). Furthermore, SP-AMS measurements of the pure and coated BC particles combined with the size distribution measurements with a SMPS provided more information about the size and morphology of the produced particles.

## 2   Experimental methods

### 2.1   Particle generation and coating

In Fig. 1, we present the schematic of the experimental setup for the generation, coating, and characterization of size-selected black carbon (BC) particles. For these experiments, we used atomized Regal black (REGAL 400R pigment black, Cabot Corp.) as a core and coated it with organics of different solubilities in water. The components selected to coat the BC particles were glutaric acid and levoglucosan, which are soluble in water, and oleic acid, which is a sparingly soluble fatty acid (Table 1). Glutaric acid is one of the products of photochemical oxidation of unsaturated hydrocarbons and fatty acids. It can also be emitted directly into the atmosphere from coal and biomass burning (Kawamura and Yasui, 2005; Li et al., 2013).



Levoglucosan is a major organic component emitted into the atmosphere from decomposition of wood during forest fires. Levoglucosan has been detected in aerosol particles at distances far from the combustion sources and is often used as an indicator for biomass burning in air quality studies (Simoneit et al., 1999). Oleic acid is a monounsaturated carboxylic acid, released to the atmosphere e.g. during meat cooking and has been used as a chemical tracer for these kind of particulate matter emissions (Rogge et al., 1991; He et al., 2004).

All of the organic components were acquired from Sigma-Aldrich and had purities higher than 99%. BC particles were produced using the atomization-drying method described in Keskinen et al. (2011). Particles were generated by an aerosol generator (Model 3076, TSI Inc., USA) from a suspension of approximately 3 g/L Regal Black (RB) in a mixture of de-ionized water (Model Maxima LS., USF Elga Ltd. with a production resistivity > 10 MΩ-cm and total organic carbon (TOC) concentration < 5 ppb) and ethanol (volume ratio 2:1). The generated polydisperse particles were passed through a custom-made silica gel diffusion drier (Fig. 1) to reach a relative humidity (RH) below 5%. Thereafter, the particles were neutralized using a radioactive $^{14}$C charge neutralizer and size-selected with a custom-made Vienna-type Differential Mobility Analyzer (DMA) (Winklmayr et al., 1991) with sample-to-sheath flow rate of 1.5 slpm to 10 slpm. The particles were coated with the chosen organic compound using a tube furnace (Vecstar, Model VCTF3). In the coating furnace, the size-selected BC particles were passed through a heated glass tube containing a vessel with the desired coating material. After first evaporating the organic vapor into the particle stream and then cooling the mixture to room temperature, the organic species condensed on the BC particles. No formation of new particles from the coating material was observed. The saturation vapor pressure of each coating compound determined the furnace temperatures needed to produce the coating. The strong temperature dependence of the saturation vapor pressure resulted in varying concentrations of organic vapor at different furnace temperatures and thus ultimately different coating thicknesses for different coating substances.

## 2.2 Particle size distribution measurements

The mobility diameter of the produced pure and coated BC particles was measured using a SMPS with sample-to-sheath flow ratio of 1.5 slpm to 10 slpm. The SMPS was composed of a Differential Mobility Analyzer (DMA) (Model 3071; TSI, Inc.) to classify particle bins according to their electrical mobilities, and a Condensation Particle Counter (CPC Model 3775; TSI, Inc.) to count the selected particles after the DMA.

## 2.3 Characterization of the particles by SP-AMS

A SP-AMS was used to investigate the chemical composition and aerodynamic size of particles containing BC and the organic coatings. The SP-AMS is a combination of the high-resolution time-of-flight aerosol mass spectrometer (AMS, Aerodyne Research Inc.) and the Single Particle Soot Photometer (SP2, Droplet Measurement Technologies Inc.). The SP-AMS has been described in detail in the literature (Onasch et al., 2012; Corbin et al., 2014; Willis et al., 2014; Ahern et al., 2016).





Briefly, in an AMS the gas phase is removed through differential pumping and the particles are focused into a narrow beam. The particles are separated by vacuum aerodynamic diameter in a particle time-of-flight region before they are vaporized, ionized and analyzed by a time-of-flight mass spectrometer (DeCarlo et al., 2006). The key difference to a standard AMS is that in addition to a tungsten vaporizer, which operates at 600°C, the instrument contains an intracavity infrared (IR) laser

module which heats refractory particles up to around 4000°C (SP mode). The tungsten vaporizer vaporizes only the non-refractory organic particulate matter and the intracavity IR laser  the refractory BC particles which do not evaporate even at 600°C (Onasch et al., 2012; Corbin et al., 2014; Willis et al., 2014; Canagaratna et al., 2015). At very high temperatures in the SP mode, also non-refractory material on the BC particles will be evaporated, but the resulting mass spectra are not comparable with the standard ones generated with the tungsten vaporizer. Thus, the two evaporation methods were alternated,

and any presented information about the BC core stems from SP mode while the information on the organic coating material was gathered from the tungsten vaporizer mode.

## 2.4    CCN activity

The CCN activity of the pure and coated BC particles was measured using a continuous flow streamwise thermal gradient

CCN counter (CCNc; Droplet Measurement Technologies Inc.) (Roberts and Nenes, 2005) (Fig. 1). In the CCNc, the size-selected aerosol was exposed to water supersaturation between 0.1% and 1.5%. Particles that activate to cloud droplets at the set supersaturation will grow large enough to be detected by the optical particle counter at the outlet of the CCNc. The ratio of the number concentration of activated particles and the total particle concentration measured with a CPC (Model 3772, TSI Inc.) yields the activated fraction. The supersaturation at which the activated fraction is 50% of the full activation is defined

as the critical supersaturation ($s_c$) for the given dry particle size. The set supersaturation was calibrated by measuring ammonium sulfate particles and comparing to theoretically calculated values.

## 3    Data analysis and theoretical methods

### 3.1    Morphology of the BC particles

The measurements with the SP-AMS and SMPS instruments provided information on the vacuum aerodynamic diameter ($d_{va}$) and the mobility diameter ($d_m$) of the pure and coated BC particles. The relationship between these two diameters can yield information about the morphology and composition of the pure and coated BC particles.

The particle effective density ($\rho_{eff}$) can be used as a measure of the non-sphericity of the particles. It is the density of the sphere with a diameter $d_m$ and the same mass of the particle in question (DeCarlo et al., 2004). The effective density of

the BC particles can be estimated using the measured diameters $d_{va}$ and $d_m$ as (Jimenez et al., 2003):





$$\rho_{eff} = \rho_0 \frac{d_{va}}{d_m} \tag{1}$$

where $\rho_0$ is unit density (1.00 g cm$^{-3}$). For spheres, the effective density is the same as particle density ($\rho_{eff} = \rho_p$), while for agglomerates $\rho_{eff} < \rho_p$ (DeCarlo et al., 2004).

The fractal dimension ($D_f$) is another parameter used to describe the geometry of agglomerated particles. For the BC particles, it is estimated by the scaling law equation for the effective density versus the mobility diameter (Rogak and Flagan, 1990; Ström et al., 1992; Virtanen et al., 2004):

$$\rho_{eff} \propto d_m^{(D_f-3)} \tag{2}$$

For spherical particles $D_f = 3$, for compact agglomerates $D_f \approx 3$, and for straight chain-like structures $D_f = 1$ (DeCarlo et al., 2004).

### 3.2   CCN activation of the pure and coated BC particles

In classical Köhler theory (Köhler, 1936; Seinfeld and Pandis, 2006), the saturation ratio of water vapor over a solution droplet of diameter $d_p$ depends on the water activity ($a_w$) (Raoult effect) and droplet curvature (Kelvin effect):

$$S = a_w \, exp \frac{4\sigma_w M_w}{RT\rho_w d_p} \tag{3}$$

where $a_w$, $\sigma_w$, $M_w$ and $\rho_w$ are the activity, surface tension, molar mass and density of the water, respectively. $R$ is the universal gas constant, $T$ is the temperature, and $d_p$ is the droplet diameter.

Köhler theory is applicable for water-soluble components like soluble organics and inorganic salts. It is not valid for insoluble or nearly insoluble compounds. Sorjamaa and Laaksonen (2007) assumed that multilayer adsorption of water molecules on the surface of the water-insoluble but wettable particles could describe cloud activation of these kind of particles.

Then the Raoult term in the Köhler theory is controlled by the adsorption of water vapor on the surface of water-insoluble particles like BC (Sorjamaa and Laaksonen, 2007; Kumar et al., 2011). Assuming equilibrium between the adsorbed water on the surface of the particles and the surrounding water vapor, the activity of water is given as (Kumar et al., 2011):

$$a_w = x_w \exp(-A\,\theta^B) \tag{4}$$

where $x_w$ is the mole fraction of the water molecules, and $A$ and $B$ are the FHH adsorption theory parameters. FHH adsorption

theory is one of the multilayer adsorption models applicable at pressures close to saturation. The parameter $A$ defines the interactions between the molecules of adsorbed components as well as between the surface and adsorbate in the first monolayer. The parameter $B$ characterizes the interactions between the surface and the adsorbed molecules in adjacent layers.



The parameters $A$ and $B$ are determined experimentally for each component. Parameter $A$ has been experimentally found to be in the range of 0.1-3.0 while, $B$ varies between 0.5-3.0 (Sorjamaa and Laaksonen, 2007). The amount of the adsorbed water is described by the surface coverage ($\theta$), which is the adsorbed number of molecules divided by the number of molecules in a monolayer (Sorjamaa and Laaksonen, 2007):

$$\theta = \frac{d_p - d_{dry}}{2d_w} \tag{5}$$

where $d_{dry}$ is the dry particle diameter and $d_w$ is the diameter of one water molecule. In the shell-and-core model introduced by Kumar et al. (2011) water vapor is assumed to be only adsorbed on the dry insoluble core and the surface coverage $\theta$ is thus given as:

$$\theta = \frac{d_p - d_{core}}{2d_w} \tag{6}$$

where $d_{core}$ is the diameter of the water-insoluble core. For water droplets generated by totally insoluble particles the mole fraction $x_w$ in Eq. 4 becomes one, but for droplets made of particles containing an insoluble core covered by a soluble coating, $x_w = 1 - x_s$ and $x_s \approx n_s / n_w$ where $n_s$ and $n_w$ are the numbers of solute and water molecules, respectively (Kumar et al., 2011). $x_s$ can then be expressed by:

$$x_s = \frac{(V_{dry} - V_{core})\rho_s v / M_s}{(V_p - V_{core})\rho_w / M_w} = \frac{\left(d_{dry}^3 - d_{core}^3\right)\rho_s v M_w}{\left(d_p^3 - d_{core}^3\right)\rho_w M_s} \tag{7}$$

where $\rho_s$ and $M_s$ are the density and molar mass of the soluble part, $V_{dry}$, $V_{core}$ and $V_p$ are the volume of the dry particle, volume of the insoluble core and volume of the drolet, respectively and $v$ is the van't Hoff factor. By assuming that the droplets are dilute solutions at the activation point, $\frac{\rho_s v M_w}{\rho_w M_s}$ becomes equal to the hygroscopicity parameter ($\kappa$) (Petters and Kreidenweis, 2007; Chang et al., 2010) and substituting Eqs. 4, 6 and 7 in Eq. 3, the saturation ratio of the coated particles will be:

$$S = \left(1 - \kappa \frac{(d_{dry}/d_{core})^3 - 1}{(d_p/d_{core})^3 - 1}\right) \exp\left(\frac{4\sigma_w M_w}{RT\rho_w d_p} - A\left(\frac{d_p - d_{core}}{2d_w}\right)^{-B}\right) \tag{8}$$

20 The supersaturation ($s$), which is equal to ($S-1$), will then become (Kumar et al., 2011):

$$s = \frac{4\sigma_w M_w}{RT\rho_w d_p} - \kappa \frac{(d_{dry}/d_{core})^3 - 1}{(d_p/d_{core})^3 - 1} - A\left(\frac{d_p - d_{core}}{2d_w}\right)^{-B} \tag{9}$$

For pure BC particles with $d_{dry} = d_{core}$, Eq.8 can be reduced to FHH adsorption activation theory (Sorjamaa and Laaksonen, 2007):

$$s = \frac{4\sigma_w M_w}{RT\rho_w d_p} - A\left(\frac{d_p - d_{dry}}{2d_w}\right)^{-B} \tag{10}$$



## 4  Results and discussion

### 4.1     Particle shape and chemical composition

Due to the agglomerated shape of the BC particles, the combination of vacuum aerodynamic ($d_{va}$) size measurements by SP-AMS and electrical mobility equivalent size ($d_m$) measurements using SMPS gives us an insight into the shape of these particles. We did the measurements for pure BC particles with the mobility sizes from 150 to 300 nm, and for BC cores with mobility diameters 150, 200 and 250 nm coated by levoglucosan, glutaric acid or oleic acid.

The combination of the SP-AMS and SMPS provided $d_{va}$ and $d_m$, which yielded the effective density of the generated particles (Eq. 1). The resulting effective densities for pure BC particles with the mobility diameters from 150 nm to 300 nm are presented in Fig. 2. The measured effective densities are in good agreements with the effective densities measured in other studies for Regal black and other BC particle types (Gysel et al., 2011, 2012; Onasch et al., 2012). The fractal dimension of the BC particles was estimated using the slopes of the curves in Fig. 2 and Eq. 2, resulting in a $D_f$  value of 2.79, which is closer to spherical or compact aggregates ($D_f = 3$) rather than chain-like structures ($D_f = 1$). The fitted $D_f$ value is also close to the value ($D_f \approx 3$) reported by Onasch et al. (2012) for the same BC type (Regal black, 400R) as we used in this study. These values suggest a compact shape of the studied BC particles. Therefore, we can assume that the organic coating only covers the BC cores like a shell. Hence, when calculating the critical supersaturations of the pure and coated BC particles using Eqs. 9 and 10, the $d_{core}$ and $d_{dry}$ were thus approximated to be the same as the mobility diameter of the BC particles before and after coating.

From SP-AMS, $d_{va}$ was derived for the total organic and BC signal separately for the coated particles (Fig. 3). The average particle sizes were slightly higher for the organic signal but generally in good agreement. Thus, the particles displayed a relatively even coating thickness, i.e. no uncoated BC particles or pure organic particles were observed.

### 4.2     Cloud activation behavior of the uncoated and coated particles

Experimentally and theoretically determined critical supersaturations of uncoated BC particles as a function of particle mobility diameter are shown in Fig. 4. Two sets of experimental data were collected under similar conditions but during two different time intervals, and the reported data are the average of these two data sets.  As is evident from Fig. 4, the critical supersaturation decreases with increasing particle diameter as expected. The parameters $A$ and $B$ of the FHH adsorption isotherm are difficult to constrain uniquely using only the CCN activation data without any additional information about the growth rate of the droplet at critical supersaturation (Dalirian et al., 2015). Nevertheless, we fitted these parameters to the two CCN data sets with the constraints $0.1 < A < 3.0$ and $0.5 < B < 3.0$ (Sorjamaa and Laaksonen, 2007). The fitted values of $A$ and $B$ for Eq. 10 were 0.5 and 1.2, respectively, and they reproduce the observations well as demonstrated in Fig. 4.





Figures 5 and 6 show the measured and theoretical critical supersaturations as a function of the total particle mobility diameter for BC core with glutaric acid and levoglucosan coating as well as pure BC and glutaric acid/levoglucosan particles. The values for the $s_c$ of the pure glutaric acid and levoglucosan were calculated using the $\kappa$-Köhler theory (Petters and Kreidenweis, 2007). The shaded areas resulted from the varying values of the $\kappa$ parameter reported in the literature for pure

glutaric acid and levoglucosan (see Table 1). As expected, the critical supersaturation is generally higher for pure BC particles than for the particles with organic coating and the pure organic particles have the lowest critical supersaturation (see Figs. 5 and 6). Furthermore, the critical supersaturation decreases when the amount of organic coating in the particles increases. As shown in the Figs. 5 and 6, the measured and calculated critical supersaturations for the coated particles are in good agreement and by increasing the amount of organic coating the $s_c$ values approach the values calculated for pure glutaric acid and

levogucosan. The same trend was observed in the study done by Hings et al. (2008) for soot particles coated by adipic acid. For the coated particles with a specific insoluble core diameter, the amount of the organic coating dominates the CCN activity of these particles and if the coating thickness is large enough compared to the initial core size, the particle will behave like pure organic.

The measured and calculated critical supersaturations as a function of the total particle mobility diameter for BC core

+ oleic acid as well as the theoretical values for the pure BC are presented in Fig. 7. Since according to the previous studies (Broekhuizen et al., 2004) and our CCNc measurements (at the measured supersaturations up to 1%), oleic acid particulate matter is CCN inactive, no hygroscopicity data is available to calculate the theoretical values for the $s_c$ of the pure oleic acid and the oleic acid coated BC particles. The CCN measurement results demonstrate that in spite of CCN inactivity of pure oleic acid, it enhances the CCN activity of the BC particles and $s_c$ of these particles decreases with increasing degree of the

oleic acid coating. One explanation for the enhancement in the CCN activation of the BC in the presence of almost insoluble oleic acid could simply be the lowering of the activation barrier to the activation by making the particle larger (Abbatt et al., 2005). However, in this case the critical supersaturation would be expected to follow the black line in Fig. 7. Another explanation would be related to the arrangement of the oleic acid molecules on the surface of the BC cores as compared with pure oleic acid particles (Garland et al., 2008) or some other mechanism through which oleic acid modifies the BC surface.

## 5   Summary and conclusions

In this study, the cloud droplet activation of uncoated and coated BC (Regal black) particles was investigated. Three kinds of organic compounds were used as coating: glutaric acid, levoglucosan and oleic acid, which can be emitted into the atmosphere from different sources e.g. biomass burning and meat cooking. Furthermore, the morphology and size of the produced particles

were investigated based on SP-AMS and SMPS measurements. In addition, the experimental CCN activity results were compared to theoretical calculations using the shell-and-core model introduced by Kumar et al. (2011).





Combining the measurements with SP-AMS and SMPS for uncoated BC particles suggested that our generated BC particles were agglomerates with fractal dimension of 2.79, which is close to spherical particles ($D_f = 3$). Therefore, we could assume that the organic coating only covered the BC cores like a shell. The coating procedure was performed for the size-selected (150, 200 and 250 nm) BC particles using a temperature-controlled tube furnace, which yielded different coating

thicknesses of the desired organic compound. For the coated BC particles, the vacuum aerodynamic diameters derived from organic and BC ion signals in the particle time-of-flight region of SP-AMS also suggested a relatively even coating of the particles. No uncoated BC particles or pure organic particles were observed after the coating procudure.

CCN activity measurements were conducted at various supersaturations, and activation ratios and critical supersaturation curves were determined for the evaluated particles. Subsequently, the experimental data for the coated BC

particles were compared to theoretical values using the shell-and-core model introduced by Kumar et al. (2011) describing CCN activation of coated insoluble components. Adsorption activation theory (e.g. Sorjamaa and Laaksonen, 2007) was used to fit the FHH adsorption parameters ($A = 0.5$ and $B = 1.2$) for the uncoated BC particles. For the BC particles coated by glutaric acid and levoglucosan, the critical supersaturations from CCN activity measurements were in good agreement with values from theoretical calculations. As expected, the coating by organics increased the CCN activity of the BC particles. The

CCN activity of the coated BC particles were governed by the fraction of organic material, and observed critical supersaturations asymptotically approached the values for pure glutaric acid and levoglucosan at larger coating thicknesses. Since the oleic acid particles were CCN inactive, no hygroscopicity data was available to calculate the $s_c$ values for the pure oleic acid. Hence no significant enhancement of CCN activity of BC by oleic acid coating was expected. However, our experimental results showed that despite the CCN inactivity of pure oleic acid, it enhanced the CCN activity of the BC particles

more than a simple size-effect would explain. One possible explanation for this behavior could be a rearrangement of the oleic acid molecules on the surface of the BC cores in such a way that the long hydrophobic alkyl chains cover the BC surface, exposing the strongly hydrophilic carboxyl groups, thus potentially providing sites to adsorb water molecules. Another possible explanation would be a decrease in surface tension due to the presence of oleic acid molecules at the interface, effectively decreasing the critical supersaturation.

Our results indicate that the amount of coating material (water-soluble or insoluble) BC particles adsorb during atmospheric aging is a parameter defining the CCN activation of these particles. We also conclude that shell-and-core model by Kumar et al. (2011), which was originally introduced for fresh dust coated by a layer of soluble salt after ageing, gives a reasonable estimate of the CCN activity of insoluble cores coated by soluble organics. Specific molecular interactions between BC and some coating molecules (like oleic acid in our case) can however yield surprising results in terms of the CCN activity of these particles. For all the coating species considered in this study, relatively thin coatings (e.g. 2 nm coating that

corresponds to about 5% of the total volume of a particle 250 nm in diameter) were enough to significantly enhance the CCN activity of the insoluble BC particles. Our results further demonstrate that BC-particles may transform in to CCN if other material is available to condense on the particles. In ambient conditions, most BC resides as internal mixtures with other





aerosol species. Hence, there is a need to include the impacts of co-emitted or later condensed soluble species in estimates of the climate impacts by BC particles.

*Acknowledgements*

5    Financial support from the Nordic Centre of Excellence CRAICC (Cryosphere-atmosphere interactions in a changing Arctic climate), Vetenskaprådet (grant n:o 2011-5120) and Knut and Alice Wallenberg foundation (Wallenberg Academy Fellowship AtmoRemove) is gratefully acknowledged.



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



**Table 1: Thermodynamic properties of components used in this study.**

|  | Molar mass (g/mol) | Density (g/cm$^3$) | Solubility in water (mol/kg) | Vapor pressure at 25ºC (Pa) | κ |
|---|---|---|---|---|---|
| Regal black |  | 1.7-1.9 [c] | Insoluble [c] |  |  |
| Glutaric acid | 132.16 [a] | 1.429 [a] | 10.8 [e] (at 25ºC) | 8.5×10$^{-4}$ [h] | 0.113-0.176 [k] |
| Levoglucosan | 162.14 [b] | 1.7±0.1 [d] | 8.23 [f] (at 20ºC) | 4.65×10$^{-5}$ [i] | 0.193-0.223 [k] |
| Oleic acid | 282.46 [a] | 0.894 [a] | Very low, 3.76×10$^{-9}$ [g] | 1.9×10$^{-6}$ [j] |  |

[a] Haynes et al. (2017)

[b] Sigma-Aldrich, Chemie GmbH

[c] Regal black 400R safety datasheet, Cabot Corp.

5   [d] Predicted, ACD/Labs Percepta Platform - PhysChem Module

[e] Soonsin et al. (2010)

[f] Zamora et al. (2011)

[g] Demond and Lindner (1993)

[h] Salo et al. (2010)

10   [i] Booth et al. (2011)

[j] Cappa et al. (2008)

[k] Petters and Kreidenweis (2007)





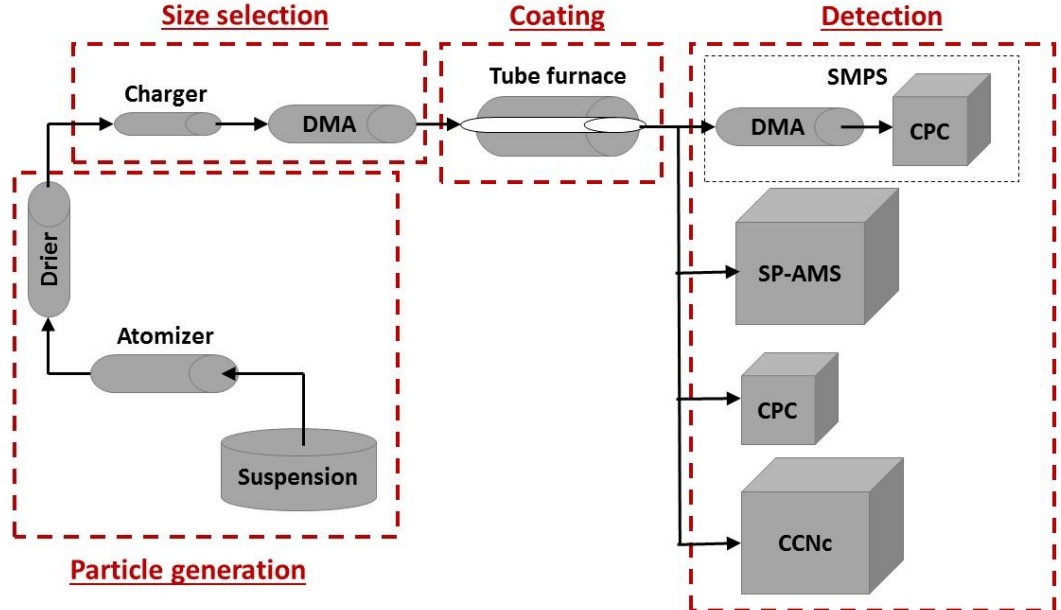

**Fig. 1: Schematic of the experimental set up, including particle generation using an atomizer and a drier, size selection with a Differential Mobility Analyzer (DMA), coating and three types of measurements: CCN activity measurements using a Condensation Particle Counter (CPC) and a Cloud Condensation Nuclei Counter (CCNc), size distribution measurements by a Scanning Mobility Particle Sizer (SMPS) and particle composition analysis by a Soot Particle Aerosol Mass Spectrometer (SP-AMS).**





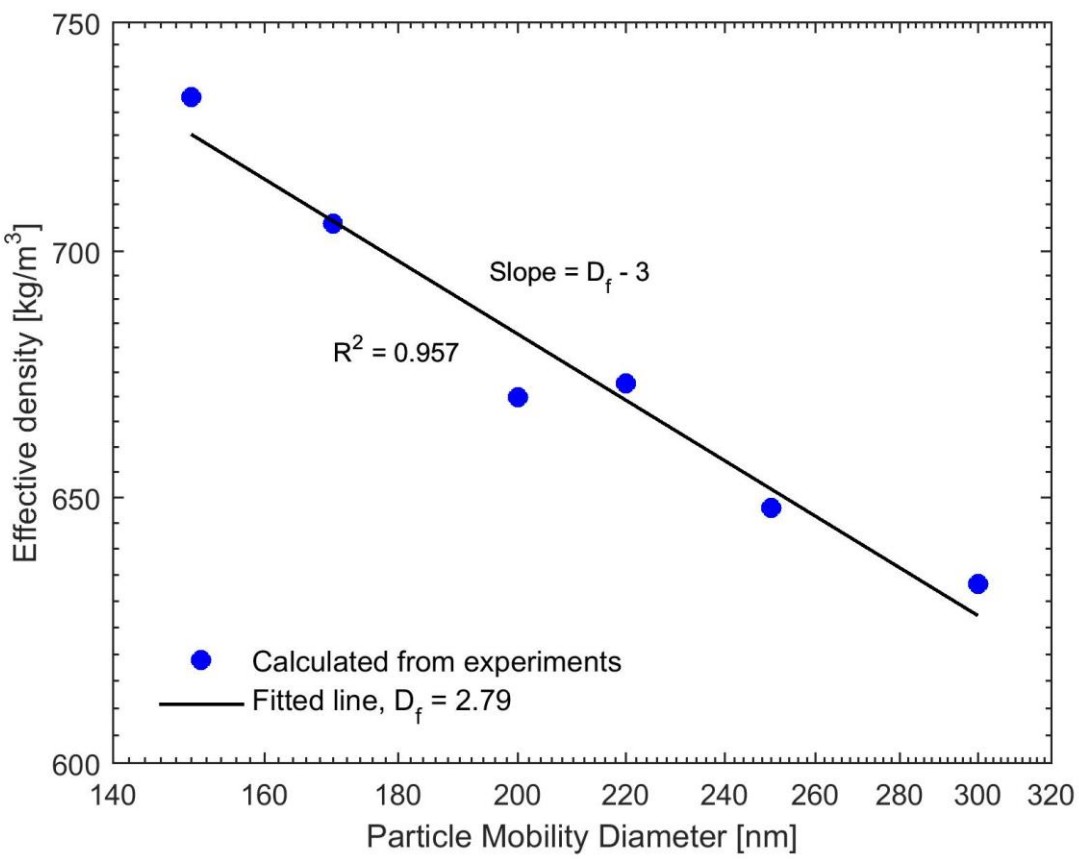

**Fig. 2: Effective density of the pure BC particles for different mobility diameters. The fitted fractal dimension is 2.79.**

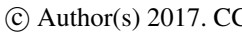



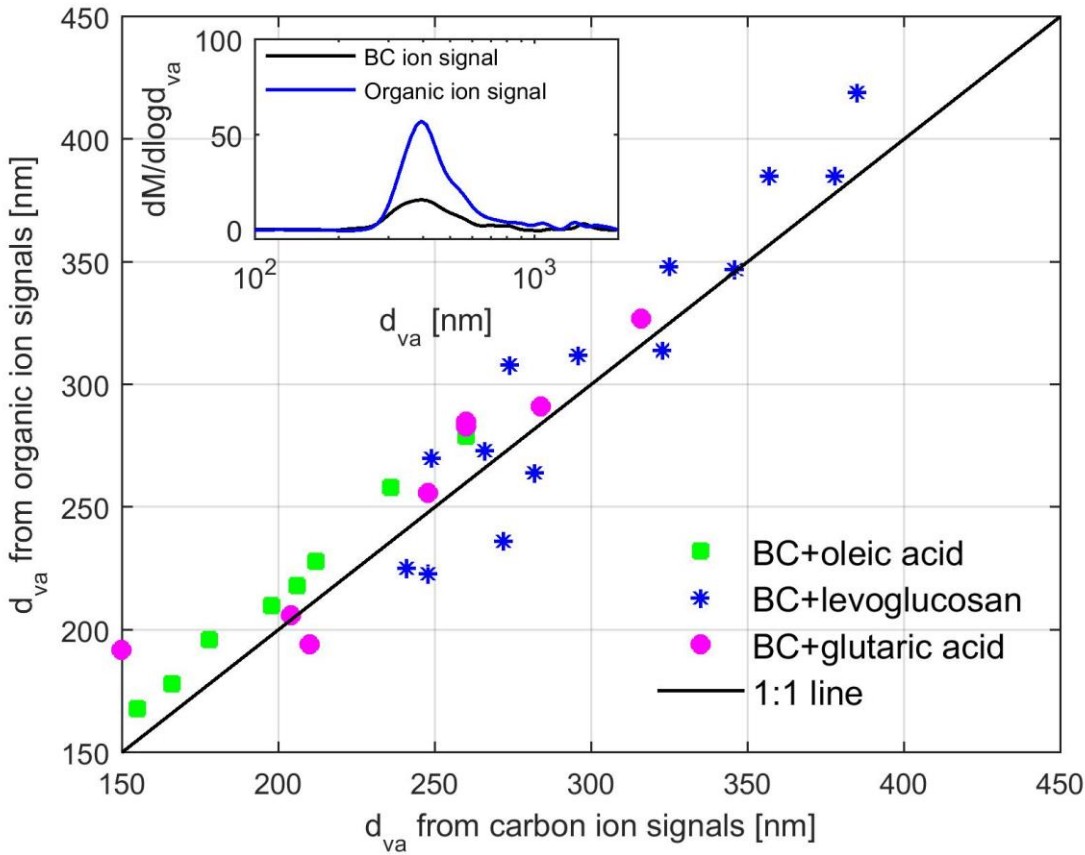

**Fig. 3: Vacuum aerodynamic diameters ($d_{va}$) from organic and carbon ions signals from SP-AMS for particles containing BC coated with various organic compounds. The inset represents mass size distribution vs. $d_{va}$ extracted from BC and organic ion signals for monodispersed 200 nm BC particles coated by 27 nm levoglucosan. The coating thickness was estimated from the mobility size measurements by SMPS.**



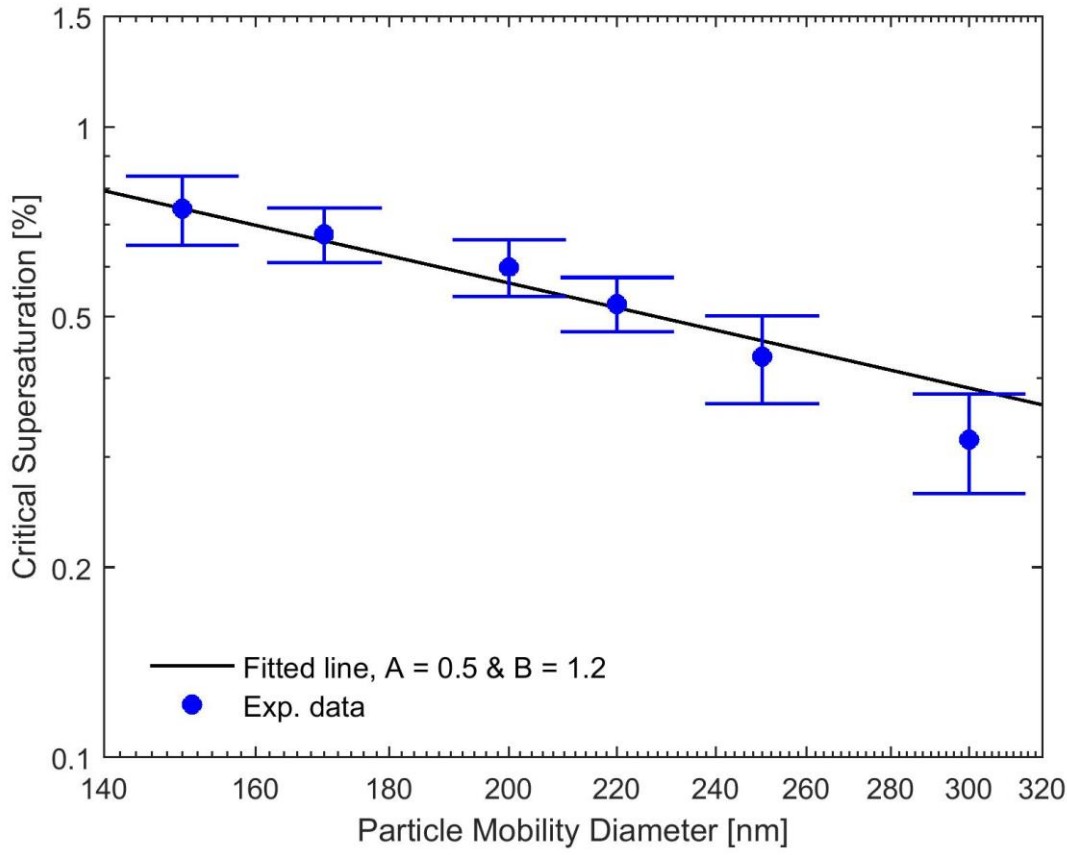

**Fig. 4: Experimental and theoretical critical supersaturations for pure BC particles for different particle mobility diameters. The theoretical curve were calculated with Eq. 10 with *A* and *B* as fitting parameters. The experimental data was the average of two sets of experiments. Error bars represent the minimum vs. maximum values of supersaturations from the two data sets.**





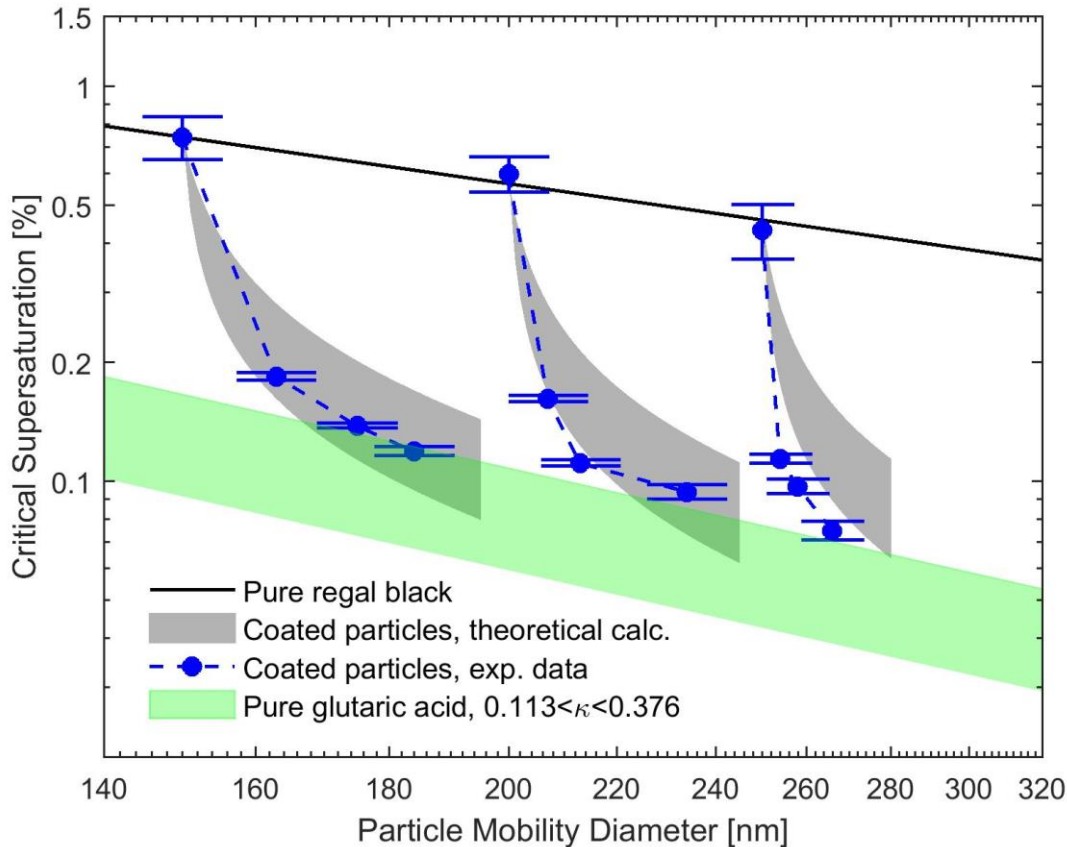

**Fig. 5: Experimental and theoretical critical supersaturations for BC particles coated by glutaric acid for different particle mobility diameters. The black line represents the calculated critical supersaturations of the pure BC particles using the FHH adsorption activation theory (Eq. 10). The gray shaded areas define the critical supersaturations calculated using Eq. 9 and variety of the $\kappa$ values reported in the literature for glutaric acid (see Table 1). The green shaded area represents the critical supersaturations from the $\kappa$-Köhler theory (Petters and Kreidenweis, 2007) for the pure glutaric acid particles with the range of $\kappa$ values from literature. Error bars represent the experimental uncertainty in the critical supersaturation $s_c$ corresponding to each mobility diameter.**





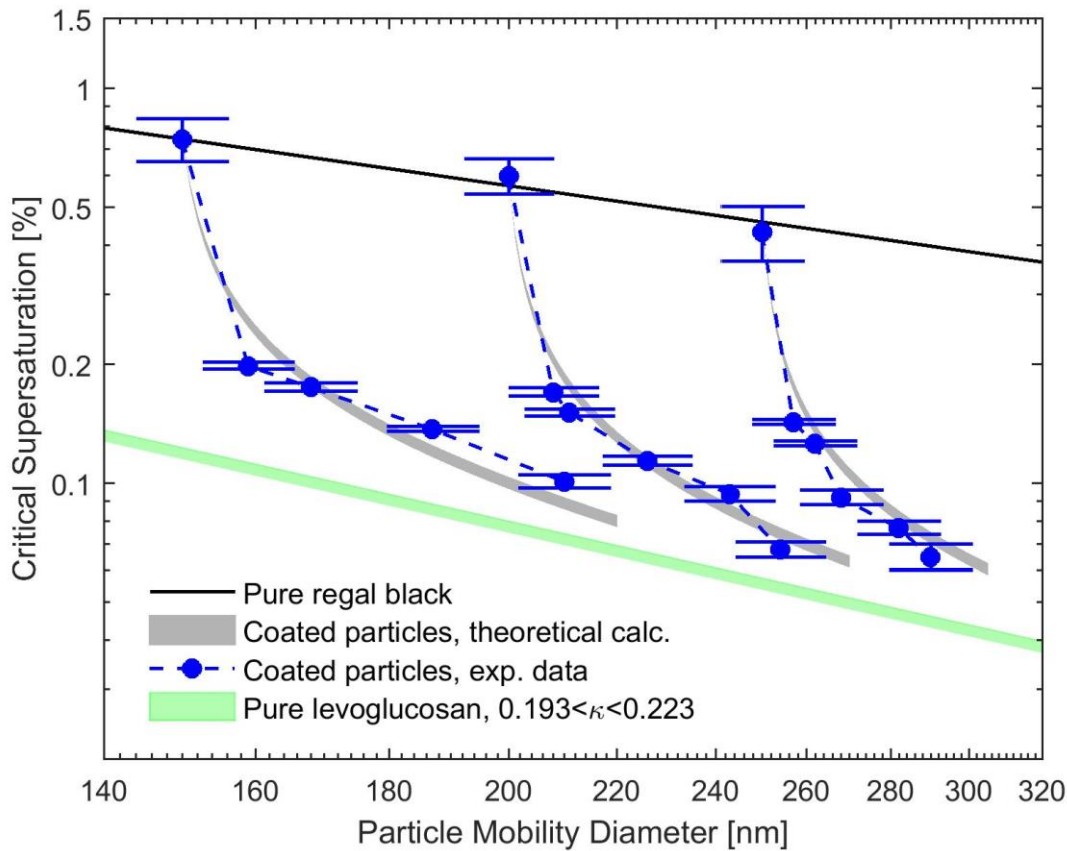

**Fig. 6: Experimental and theoretical critical supersaturations for BC particles coated by levoglucosan for different particle mobility diameters. The black line represents the calculated critical supersaturations of the pure BC particles using the FHH adsorption activation theory (Eq. 10). The gray shaded areas define the critical supersaturations calculated using Eq. 9 and variety of the $\kappa$ values reported in the literature for levoglucosan (see Table 1). The green shaded area represents the critical supersaturations from the $\kappa$-Köhler theory (Petters and Kreidenweis, 2007) for the pure levoglucosan particles with the range of $\kappa$ values from literature. Error bars represent the experimental uncertainty in the critical supersaturation $s_c$ corresponding to each mobility diameter.**



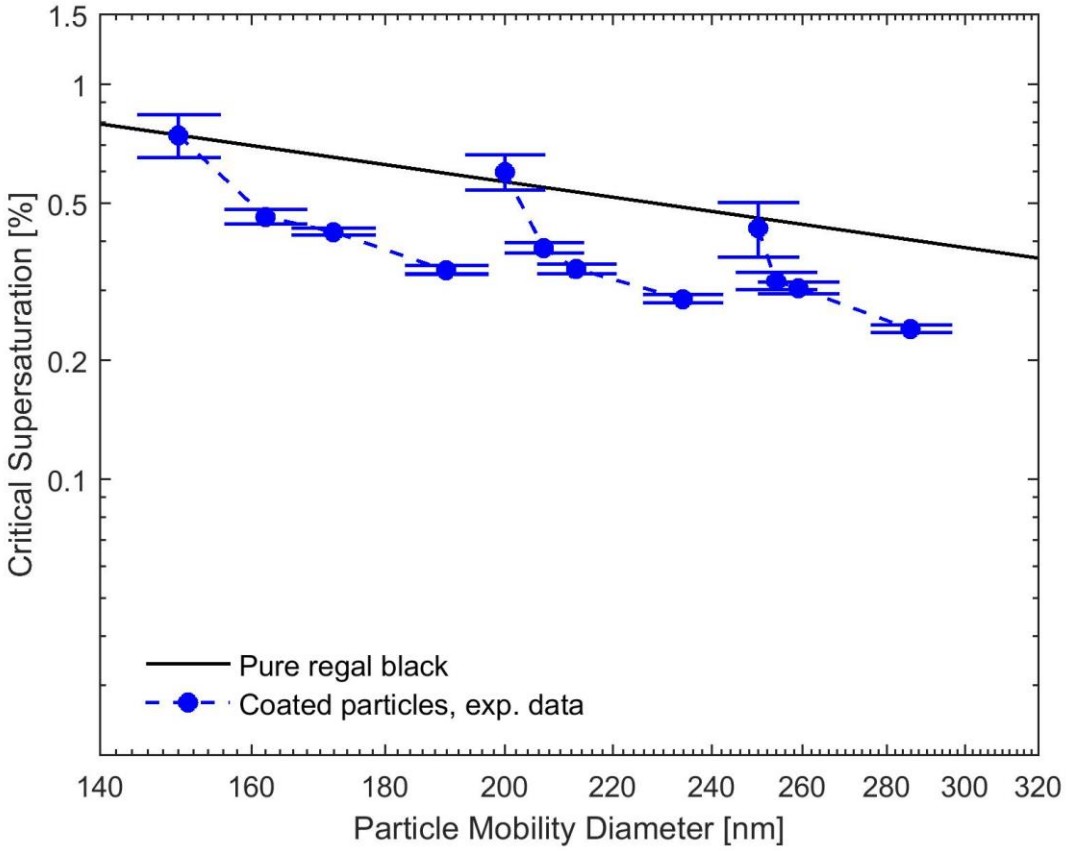

**Fig. 7: Experimental and theoretical critical supersaturations for coated BC by oleic acid for different particle mobility diameters. Black line represents the calculated critical supersaturations of the pure BC particles using FHH adsorption activation theory (Eq. 10). No CCN activation was observed for pure oleic acid particles.**

