# Peer review of "Cloud droplet activation of black carbon particles coated with organic compounds of varying solubility"

_Atmospheric Chemistry and Physics, 2017_

## Referee Comment (RC1) · Anonymous Referee #1 · 9 Feb 2018

General comments

The topic discussed in this paper is very important, and of interest to the community and the ACP readership. I found the paper to be very clear and well written, and the study to be carefully conducted in their experiments as well as in the analysis and interpretation of the results. Therefore, I suggest publication with only minor suggestions. I have a couple of general comments:

1. The BC particles were produced from a liquid suspension using an atomizer and a dryer before coating, this changed the morphological structure of the BC to a compact BC structure, as the authors discuss. It would help mentioning that the atmospheric

[Figure]

BC can have different degrees of compaction (there are a few studies showing this for laboratory, but also for atmospheric particles), and discussing briefly how the presence of more open-structured BC particles in the atmosphere might affect the conclusions of this study, and the models applied. For example, that might be relevant for fresh vs. aged BC particles. Related to this topic, a lot of work published in the literature on BC morphology, compaction, and coating is completely neglected here (in particular several electron microscopy, SP2, and optical studies from different groups around the world). I think that mentioning a few of these studies would improve the paper.

2. The BC particles were size selected with a DMA before being coated. This maintains the core constant while the coating thickness is increased. This approach is fine for the most part and produces interesting results. However, to untangle the effect of the coating hygroscopicity from that of size, it would have been interesting to size-select before and after the coating stage as well, to maintain the overall particle size constant, while changing the coating thickness; in this way, isolating the effect of the overall particle size (this would be particularly interesting for the case of the oleic acid). I am not suggesting the authors should conduct such experiments for this manuscript, as I think the results of the current study are very interesting on their own, but they could briefly discuss this possibility for future studies.

Specific comments

- Page 2 line 31, there are several studies, including some recent, that analyze and quantify the effect of coating, mixing and compaction for BC with the detail and unambiguity of electron microscopy, as well. Studies are available for both laboratory, as well as, ambient BC particles. It might be worth discussing some here.

- Page 3, line7, "of uncoated" seems not to belong here considering they are talking about multilayer models.

- Page 3, line 27. Please provide a sentence or two on why regal back and how does that represent (how well it acts as) a surrogate for atmospheric BC. This in addition to

the compaction issue mentioned in the general comments section.

- Figure 5: the theoretical calculation seems to perform less well for the larger core diameter; in fact, the range of the gray band does not seem to intersect with the experimental data even considering their uncertainties. Is there any reason for that? A short discussion would be interesting.

- Figure 6, maybe this was mentioned and I missed it, but why are the theoretical calculations so much narrower here than in figure 5? I guess the spread reflects directly the k range for the two coating materials, narrower for levoglucosan than for glutaric acid. Can the authors comment on that?

Technical corrections

- Page 3, line 13, consider adding the article "the" in front of Soot Particle Aerosol Mass Spectrometer

- Page 10, line 7, "procedure", should be "procedure", probably.

---

## Referee Comment (RC2) · Anonymous Referee #2 · 18 Mar 2018

By reviewing the manuscript on the CCN activation properties of BC particles coated with a few organic species I feel myself rather uncomfortable. The study is a carefully planned and executed combination of experimental work and theoretical calculations aiming at providing new insights into the CCN behavior of atmospheric BC particles. However, the entire approach looks as a textbook-like routine exercise that contains no traces of novelty and innovation that would have been required by a high-standard journal like Atmospheric Chemistry and Physics. Any laboratory study in itself is free to use virtually any combinations of agents and conditions, yet it would be expected to be a quasi-realistic model of physical reality. The basic concept of the present study does not fulfil this fundamental requirement. What sort of real-life BC particles do Regal

black stand for? Aged diesel soot particles or BC particles from flaming biomass combustion? Levoglucosan, which is an abundant pyrolysis product of wood combustion, is not a semi-volatile species that is available for adsorption or condensation in the global atmosphere such as PAHs or n-alkanes. It is always present internally mixed with smoke particles, not as a gaseous species. Oleic acid is also a primary tracer which–unlike levoglucosan–is present in the gas phase but on a very limited spatial scale near its sources. I would doubt that this photochemically reactive species can make it to the free troposphere to participate in cloud nucleation. I wonder if anybody has ever detected oleic acid in cloud water or precipitation. In spite of these serious limitations oleic acid experiments at least yielded some unexpected results which are not fully exploited in the manuscript. Perhaps a molecular adsorption modelling approach would have helped explain the observations. Glutaric acid does exist as SOA product in the atmosphere, though at far lower concentrations than smaller dicarboxylic acids or other SVOC species. In addition, a very similar study was published for the CCN effect of adipic acid. I suspect that upon releasing fresh BC particles from any source there is a plethora of co-emitted semi-volatile species that are ready to be adsorbed onto their surfaces. It is strange that in the experimental section no temperature values are given for the coating procedure. These temperatures would also indicate that the atmospheric occurrence of such processes is unlikely. For lack of originality, the manuscript just declares plain trivialities such as on Page 9 Line 5-6 "As expected, the critical supersaturation is generally higher for pure BC particles than for the particles with organic coating and the pure organic particles have the lowest critical supersaturation". Overall, this manuscript presents a lab-based approach in combination with a well-established theoretical approach that has little if any atmospheric relevance. It is a textbook-like repetition of previous studies and completely lacks originality and innovation.

---

## Author Comment (AC1) · 3 May 2018

**Reviewer #1:**

*We thank Reviewer #1 for positive, constructive and detailed comments, which we will account for in the revised manuscript. Our point-by-point responses to the issues raised by the reviewer are below.*

**General comments**

1. The BC particles were produced from a liquid suspension using an atomizer and a dryer before coating, this changed the morphological structure of the BC to a compact BC structure, as the authors discuss. It would help mentioning that the atmospheric BC can have different degrees of compaction (there are a few studies showing this for laboratory, but also for atmospheric particles), and discussing briefly how the presence of more open-structured BC particles in the atmosphere might affect the conclusions of this study, and the models applied. For example, that might be relevant for fresh vs. aged BC particles. Related to this topic, a lot of work published in the literature on BC morphology, compaction, and coating is completely neglected here (in particular several electron microscopy, SP2, and optical studies from different groups around the world). I think that mentioning a few of these studies would improve the paper.

*We agree with the referee and will add the following text to the revised manuscript and will modify the last paragraph in page 2:*

*"Laboratory measurements have indicated that by increasing the amount of the coating on the BC particles, the dynamic shape factor of these particles decreases, and fractal BC aggregates become restructured and more compact (Saathoff et al. 2003; Slowik et al. 2007; Zhang et al. 2008; Pagels et al. 2009; Tritscher et al. 2011; Ghazi and Olfert 2012). Investigating ambient BC particles have shown that BC particles coated by secondary aerosol constituents during atmospheric aging transform from a fractal to spherical and further fully compact shapes (Peng et al. 2016; Zhang et al. 2016). Furthermore, ambient BC measurements have demonstrated that aging of BC particles and coating by other material via condensation and coagulation can enhance the light absorption capability of BC particle (Khalizov et al. 2009; Moffet and Prather 2009; Chan et al. 2011; Liu et al. 2015; Zhang et al. 2017; Xu et al. 2018). Although this enhancement of light adsorption properties of BC-containing particles is still a large uncertainty in modelling direct radiative forcing of BC particles. Furthermore, there are uncertainties in modelling the indirect radiative forcing of the BC-containing particles, due to e.g. lack of knowledge about cloud interactions of BC-containing particles and the role of the co-emitted species. To overcome ...."*

*We will also add a brief discussion of the representativity of our BC particles to the revised manuscript.*

2. The BC particles were size selected with a DMA before being coated. This maintains the core constant while the coating thickness is increased. This approach is fine for the most part and produces interesting results. However, to untangle the effect of the coating hygroscopicity from that of size, it would have been interesting to size select before and after

the coating stage as well, to maintain the overall particle size constant, while changing the coating thickness; in this way, isolating the effect of the overall particle size (this would be particularly interesting for the case of the oleic acid). I am not suggesting the authors should conduct such experiments for this manuscript, as I think the results of the current study are very interesting on their own, but they could briefly discuss this possibility for future studies.

*Indeed, these kind of experiments would be an interesting topic for a future study. We will add a statement to the revised manuscript mentioning this possibility for future studies. However, they would be somewhat challenging (although perhaps not impossible) with the present setup, where the temperature of the furnace in effect determines the coating thickness. We selected the size of the BC cores first by a DMA and the coating thickness was varied by changing the furnace temperature. We then measured the size distribution of the coated particles exiting the furnace. At any given temperature, we then estimated the size of the coated particles using the peak value from the size distribution curves.*

**Specific comments**

- Page 2 line 31, there are several studies, including some recent, that analyze and quantify the effect of coating, mixing and compaction for BC with the detail and unambiguity of electron microscopy, as well. Studies are available for both laboratory, as well as, ambient BC particles. It might be worth discussing some here.

*We will modify the revised manuscript by changing the sentence in question to (see also our response to general comment 1):*

*Laboratory measurements have indicated that by increasing the amount of the coating on the BC particles, the dynamic shape factor of these particles decreases, fractal BC aggregates become restructured and more compact (Saathoff et al. 2003; Slowik et al. 2007; Zhang et al. 2008; Pagels et al. 2009; Tritscher et al. 2011; Ghazi and Olfert 2012). Investigating ambient BC particles have shown that BC particles coated by secondary aerosol constituents during atmospheric aging transform from a fractal to spherical and further fully compact shapes (Peng et al. 2016; Zhang et al. 2016).*

- Page 3, line7, "of uncoated" seems not to belong here considering they are talking about multilayer models.

*The reviewer is correct. We will remove the reference to the uncoated particles from the first sentence of the paragraph. The adsorption activation model described by Sorjamaa & Laaksonen (2007) assumes that the CCN activation of insoluble but wettable compounds happens through multilayer adsorption of water molecules. This model was developed later by Kumar et al. (2011) to include the CCN activation of the insoluble particles coated by soluble salts.*

- Page 3, line 27. Please provide a sentence or two on why regal back and how does that represent (how well it acts as) a surrogate for atmospheric BC. This in addition to the compaction issue mentioned in the general comments section.

*We will add the following text to the revised manuscript:*

*Regal Black (Cabot REGAL R400 pigment black), which was provided by Cabot Corp., USA, is a surrogate for collapsed soot (Sedlacek et al. 2015) and is the recommended calibration standard for the SP-AMS (Onasch et al., 2012). This compound has been used in different studies (Onasch et al. 2012; Corbin et al. 2014; Healy et al. 2015; Sedlacek et al. 2015) as a model of refractory carbonaceous compounds to estimate the chemical and physical properties of the black carbon particles and Canagaratna et al. (2015) have shown that regal black and flame soot appear very similar, at least from the perspective of mass spectrometry. However, it should be borne in mind that in the ambient BC particles can vary significantly in terms of their physical and chemical properties, and is usually mixed with other pollutants present in the atmosphere.*

*This clarification will be added to the manuscript.*

- Figure 5: the theoretical calculation seems to perform less well for the larger core diameter; in fact, the range of the grey band does not seem to intersect with the experimental data even considering their uncertainties. Is there any reason for that? A short discussion would be interesting.

*This is a very good question, to which we do not have a definite answer. One reason might be just a larger uncertainty in the κ values than what is considered in the calculations. Different values have been reported for κ in different studies (Petters and Kreidenweis 2007; Chan et al. 2008; Petters et al. 2016). The κ we used for glutaric acid is from (Petters and Kreidenweis 2007) and is between 0.113-0.376 (we will correct it in the table 1).*

*We will add this clarification to the manuscript.*

- Figure 6, maybe this was mentioned and I missed it, but why are the theoretical calculations so much narrower here than in figure 5? I guess the spread reflects directly the k range for the two coating materials, narrower for levoglucosan than for glutaric acid. Can the authors comment on that?

*This is correct and we have thus added the following sentence to our manuscript, first paragraph on page 9:*

*"The theoretical calculations of $s_c$ are narrower for levoglucosan compared to glutaric acid, because the κ range is narrower for levoglucosan (it is between 0.193-0.223 for levoglucosan and 0.113-0.376 (we will correct it in the table 1) for glutaric acid)."*

**Technical corrections**

- Page 3, line 13, consider adding the article "the" in front of Soot Particle Aerosol Mass Spectrometer

*We will add "the" in front of Soot Particle Aerosol Mass Spectrometer.*

- Page 10, line 7, "procudure", should be "procedure", probably.

*We will correct the word "procedure".*

***References***

Canagaratna MR, Massoli P, Browne EC, et al (2015) Chemical compositions of black carbon particle cores and coatings via soot particle aerosol mass spectrometry with photoionization and electron ionization. J Phys Chem A 119:4589–4599 . doi: 10.1021/jp510711u

Chan MN, Kreidenweis SM, Chan CK (2008) Measurements of the hygroscopic and deliquescence properties of organic compounds of different solubilities in water rand their relationship with cloud condensation nuclei activities. Environ Sci Technol 42:3602–3608

Chan TW, Brook JR, Smallwood GJ, Lu G (2011) Time-resolved measurements of black carbon light absorption enhancement in urban and near-urban locations of southern Ontario, Canada. Atmos Chem Phys 11:10407–10432 . doi: 10.5194/acp-11-10407-2011

Corbin JC, Sierau B, Gysel M, et al (2014) Mass spectrometry of refractory black carbon particles from six sources: Carbon-cluster and oxygenated ions. Atmos Chem Phys 14:2591–2603 . doi: 10.5194/acp-14-2591-2014

Ghazi R, Olfert JS (2012) Coating Mass Dependence of Soot Aggregate Restructuring Due to Coatings of Oleic Acid and Dioctyl Sebacate. Aerosol Sci Technol 6826:121023074234007 . doi: 10.1080/02786826.2012.741273

Healy RM, Wang JM, Jeong C, et al (2015) Light-absorbing properties of ambient black carbon and brown carbon from fossil fuel and biomass burning sources. J Geophys Res 6619–6633 . doi: 10.1002/2015JD023382.Received

Khalizov AF, Xue H, Wang L, et al (2009) Enhanced Light Absorption and Scattering by Carbon Soot Aerosol Internally Mixed with Sulfuric Acid Enhanced Light Absorption and Scattering by Carbon Soot Aerosol Internally Mixed with Sulfuric Acid. 1066–1074 . doi: 10.1021/jp807531n

Kumar P, Sokolik IN, Nenes  a. (2011) Cloud condensation nuclei activity and droplet activation kinetics of wet processed regional dust samples and minerals. Atmos Chem Phys 11:8661–8676 . doi: 10.5194/acp-11-8661-2011

Liu S, Aiken AC, Gorkowski K, et al (2015) Enhanced light absorption by mixed source black and brown carbon particles in UK winter. Nat Commun 6: . doi: 10.1038/ncomms9435

Moffet RC, Prather KA (2009) In-situ measurements of the mixing state and optical properties of soot with implications for radiative forcing estimates. Proc Natl Acad Sci 106:11872–11877 . doi: 10.1073/pnas.0900040106

Onasch TB, Trimborn A, Fortner EC, et al (2012) Soot Particle Aerosol Mass Spectrometer: Development, Validation, and Initial Application. Aerosol Sci Technol 46:804–817 . doi: 10.1080/02786826.2012.663948

Pagels J, Khalizov AF, McMurry PH, Zhang RY (2009) Processing of Soot by Controlled Sulphuric Acid and Water Condensation—Mass and Mobility Relationship. Aerosol Sci Technol 43:629–640 . doi: 10.1080/02786820902810685

Peng J, Hu M, Guo S, et al (2016) Markedly enhanced absorption and direct radiative forcing of black carbon under polluted urban environments. Proc Natl Acad Sci 113:4266–4271 . doi: 10.1073/pnas.1602310113

Petters MD, Kreidenweis SM (2007) A single parameter representation of hygroscopic growth and cloud condensation nucleus activity. Atmos Chem Phys 7:1961–1971 . doi: 10.5194/acp-7-1961-2007

Petters MD, Kreidenweis SM, Ziemann PJ (2016) Prediction of cloud condensation nuclei activity for organic compounds using functional group contribution methods. Geosci Model Dev 9:111–124 . doi: 10.5194/gmd-9-111-2016

Saathoff H, Naumann K-H, Schnaiter M, et al (2003) Coating of soot and (NH4)2SO4 particles by ozonolysis products of α-pinene. J Aerosol Sci 34:1297–1321 . doi: 10.1016/S0021-8502(03)00364-1

Sedlacek AJ, Lewis ER, Onasch TB, et al (2015) Investigation of Refractory Black Carbon-Containing Particle Morphologies Using the Single-Particle Soot Photometer (SP2). Aerosol Sci Technol 49:872–885 . doi: 10.1080/02786826.2015.1074978

Slowik JG, Cross ES, Han J-H, et al (2007) Measurements of Morphology Changes of Fractal Soot Particles using Coating and Denuding Experiments: Implications for Optical Absorption and Atmospheric Lifetime. Aerosol Sci Technol 41:734–750 . doi: 10.1080/02786820701432632

Tritscher T, Juanyi Z, Martin M, et al (2011) Changes of hygroscopicity and morphology during ageing of diesel soot. Environ Res Lett 6: . doi: 10.1088/1748-9326/6/3/034026

Xu X, Zhao W, Qian X, et al (2018) Influence of photochemical aging on light absorption of atmospheric black carbon and aerosol single scattering albedo. 2:1–29 . doi: 10.5194/acp-2018-59

Zhang R, Khalizov AF, Pagels J, et al (2008) Variability in morphology , hygroscopicity , and optical properties of soot aerosols during atmospheric processing. 105:10291–10296

Zhang Y, Zhang Q, Cheng Y, et al (2016) Measuring the morphology and density of internally mixed black carbon with SP2 and VTDMA: New insight into the absorption enhancement of black carbon in the atmosphere. Atmos Meas Tech 9:1833–1843 . doi: 10.5194/amt-9-1833-2016

Zhang Y, Zhang Q, Cheng Y, et al (2017) Amplification of light absorption of black carbon associated with air. Atmos Chem Phys Discuss 1–27 . doi: 10.5194/acp-2017-983

---

## Author Comment (AC2) · 3 May 2018

**Reviewer #2:**

*We thank Reviewer #2 for taking the time to consider and review our manuscript. We respectfully disagree, however, with the reviewer on the purpose and value of the presented work. We feel there is a fundamental difference of opinion in 1) what can be considered as "text-book knowledge"; 2) the value of well-defined laboratory studies with simple model systems for evaluating theoretical approaches used to describe more complex atmospheric systems. Our point-by-point responses to the issues raised by the reviewer are below.*

By reviewing the manuscript on the CCN activation properties of BC particles coated with a few organic species I feel myself rather uncomfortable. The study is a carefully planned and executed combination of experimental work and theoretical calculations aiming at providing new insights into the CCN behavior of atmospheric BC particles. However, the entire approach looks as a textbook-like routine exercise that contains no traces of novelty and innovation that would have been required by a high-standard journal like Atmospheric Chemistry and Physics. Any laboratory study in itself is free to use virtually any combinations of agents and conditions, yet it would be expected to be a quasi-realistic model of physical reality. The basic concept of the present study does not fulfil this fundamental requirement.

*We are sorry to hear that the reviewer feels uncomfortable, but thank him/her for acknowledging the quality of the presented work nevertheless. However, there seems to be a fundamental misunderstanding regarding the purpose and novelty of our work. As we wrote in the "Introduction", the CCN activation of uncoated and coated insoluble particles, such as BC-particles coated with soluble species, is usually described theoretically by multilayer adsorption models accounting for the curvature of the particles. One of these theories is adsorption activation theory which is a combination of FHH adsorption isotherms and classical Köhler theory to describe the equilibrium growth of insoluble particles. Later, Kumar et al. (2011) introduced a new framework of CCN activation of dust containing a soluble salt fraction, based on a combination of the classical Köhler and FHH adsorption theories. However, systematic experimental testing of the applicability of combined Köhler and FHH theory with agglomerated insoluble particles coated with organic species of varying solubility is lacking. Unlike what the reviewer claims, these theoretical approaches are not "well-established" in terms of systematically testing their applicability with known molecular species. This was the main motivation and novelty of our study and we will highlight this now more clearly in the Introduction of the revised manuscript.*

What sort of real-life BC particles do Regal black stand for? Aged diesel soot particles or BC particles from flaming biomass combustion? Levoglucosan, which is an abundant pyrolysis product of wood combustion, is not a semi-volatile species that is available for adsorption or condensation in the global atmosphere such as PAHs or n-alkanes. It is always present internally mixed with smoke particles, not as a gaseous species. Oleic acid is also a primary tracer which–unlike levoglucosan–is present in the gas phase but on a very limited spatial scale near its sources. I would doubt that this photochemically reactive species can make it to the free troposphere to participate in cloud nucleation. I wonder if anybody has ever detected oleic acid in cloud water or precipitation. In spite of these serious limitations oleic acid experiments at least yielded some unexpected results which are not fully exploited in the manuscript. Perhaps a molecular adsorption modelling approach would have helped explain the

observations. Glutaric acid does exist as SOA product in the atmosphere, though at far lower concentrations than smaller dicarboxylic acids or other SVOC species. In addition, a very similar study was published for the CCN effect of adipic acid. I suspect that upon releasing fresh BC particles from any source there is a plethora of co-emitted semi-volatile species that are ready to be adsorbed onto their surfaces. It is strange that in the experimental section no temperature values are given for the coating procedure. These temperatures would also indicate that the atmospheric occurrence of such processes is unlikely.

*We are well-aware that in the atmosphere a myriad of different organic and inorganic compounds accompany BC in the particulate phase. The main purpose of this study was to test the model frameworks used to describe these extremely complex mixtures in various atmospheric models by comparing their predictions to well-defined particles generated in the laboratory. It is as well-established approach to use laboratory measurements of model compounds representing the variation of properties relevant for the studied processes to evaluate theoretical frameworks. In fact, a major part of laboratory measurements represent this kind of approach. Hence we feel that the criticism raised by the reviewer is poorly justified and unfair.*

*As also pointed out in our responses to Reviewer #1, Regal black has been standardly used as a surrogate for collapsed soot (Sedlacek et al. 2015) and is the recommended calibration standard for the SP-AMS (Onasch et al., 2012). This compound has been used in different studies (Onasch et al. 2012; Corbin et al. 2014; Healy et al. 2015; Sedlacek et al. 2015) as a model of refractory carbonaceous compounds to estimate the chemical and physical properties of the black carbon particles. Canagaratna et al. (2015) have shown that regal black and flame soot appear very similar, at least from the perspective of the mass spectrometry. However, it should of course be borne in mind that in the ambient BC particles can vary significantly in terms of their physical and chemical properties, and is usually mixed with other pollutants present in the atmosphere. We will highlight this in the revised manuscript (see also response to Reviewer #1).*

*We would also like to highlight that the studied organic substances were chosen based on their properties, not solely based on their atmospheric relevance. The solubility and other properties of atmospheric organic material varies considerably (e.g. Goldstein and Galbally 2007; Jimenez et al. 2009) and this variation directed our choice. We will, however, add a table detailing the coating temperatures to the revised manuscript. The oleic acid results simply highlight the need of laboratory measurements with simple model compounds. Our results show that by using the existing model frameworks, we cannot explain all of the experimental observations, but more theoretical work is needed. We will highlight this in the revised manuscript.*

For lack of originality, the manuscript just declares plain trivialities such as on Page 9 Line 5-6 "As expected, the critical supersaturation is generally higher for pure BC particles than for the particles with organic coating and the pure organic particles have the lowest critical supersaturation". Overall, this manuscript presents a lab-based approach in combination with a well-established theoretical approach that has little if any atmospheric relevance. It is a textbook-like repetition of previous studies and completely lacks originality and innovation.

*First, we would like to point out that the fact that the present theories are shown to work well for two of the studied organic compounds does not mean that the results are not novel. Not all novel results need to be surprising or worrying in terms of the application of the theory for atmospherically relevant calculations. The reviewer claims that our study is a repetition of previous studies, but does not provide any references to back up these claims. We are not aware of any previous studies that investigate the applicability of the adsorption-activation approaches for BC particles systematically coated with the studied organic compounds using a similar approach to the one presented here. We therefore find the criticism unjustified.*

**References**

Canagaratna MR, Massoli P, Browne EC, et al (2015) Chemical compositions of black carbon particle cores and coatings via soot particle aerosol mass spectrometry with photoionization and electron ionization. J Phys Chem A 119:4589–4599 . doi: 10.1021/jp510711u

Corbin JC, Sierau B, Gysel M, et al (2014) Mass spectrometry of refractory black carbon particles from six sources: Carbon-cluster and oxygenated ions. Atmos Chem Phys 14:2591–2603 . doi: 10.5194/acp-14-2591-2014

Goldstein AH, Galbally IE (2007) Known and unexplored organic constituents in the earth's atmosphere. Environ Sci Technol 41:1514–1521 . doi: 10.1021/es072476p

Healy RM, Wang JM, Jeong C, et al (2015) Light-absorbing properties of ambient black carbon and brown carbon from fossil fuel and biomass burning sources. J Geophys Res 6619–6633 . doi: 10.1002/2015JD023382.Received

Jimenez JL, Canagaratna MR, Donahue NM, et al (2009) Evolution of organic aerosols in the atmosphere. Science (80- ) 326:1525–1529 . doi: 10.1126/science.1180353

Onasch TB, Trimborn A, Fortner EC, et al (2012) Soot Particle Aerosol Mass Spectrometer: Development, Validation, and Initial Application. Aerosol Sci Technol 46:804–817 . doi: 10.1080/02786826.2012.663948

Sedlacek AJ, Lewis ER, Onasch TB, et al (2015) Investigation of Refractory Black Carbon-Containing Particle Morphologies Using the Single-Particle Soot Photometer (SP2). Aerosol Sci Technol 49:872–885 . doi: 10.1080/02786826.2015.1074978

---

## Author Response (AR1)

**Dear Editor,**

5

30

Thank you for considering our manuscript. We would also like to thank both reviewers for the time they have spent on our manuscript and their contributions to improving it. Please find below our point-by-point responses to the reviewer comments along with the revised manuscript. The reviewer comments are written in normal font, and our responses in italics.

**Reviewer #1:**

10 We thank Reviewer #1 for positive, constructive and detailed comments, which we will account for in the revised manuscript. Our point-by-point responses to the issues raised by the reviewer are below.

**General comments**

The BC particles were produced from a liquid suspension using an atomizer and a dryer before coating, this changed the morphological structure of the BC to a compact BC structure, as the authors discuss. It would help mentioning that the atmospheric BC can have different degrees of compaction (there are a few studies showing this for laboratory, but also for atmospheric particles), and discussing briefly how the presence of more open-structured BC particles in the atmosphere might affect the conclusions of this study, and the models applied. For example, that might be relevant for fresh vs. aged BC particles. Related to this topic, a lot of work published in the literature on BC morphology, compaction, and coating is completely neglected here (in particular several electron microscopy, SP2, and optical studies from different groups around the world). I think that mentioning a few of these studies would improve the paper.

We agree with the referee and have added the following text to the beginning of the last paragraph of page 2 in the revised manuscript:

"The structure and properties of BC particles in the atmosphere can vary considerably. Laboratory measurements have indicated that by increasing the amount of the coating on the BC particles, the dynamic shape factor of these particles decreases, and fractal BC aggregates become restructured and more compact (Saathoff et al. 2003; Slowik et al. 2007; Zhang et al. 2008; Pagels et al. 2009; Tritscher et al. 2011; Ghazi and Olfert 2012). Investigating ambient BC particles have shown that BC particles coated by secondary aerosol constituents during atmospheric aging transform from a fractal to spherical and further fully compact shapes (Peng et al. 2016; Zhang et al. 2016). Furthermore, ambient BC measurements have demonstrated that aging of BC particles and coating by other material via

condensation and coagulation can enhance the light absorption capability of BC particles (Khalizov et al. 2009; Moffet and Prather 2009; Chan et al. 2011; Liu et al. 2015; Zhang et al. 2017; Xu et al. 2018). This enhancement of light adsorption properties of BC-containing particles is, however, still a large uncertainty in modelling the direct radiative forcing of BC. In addition, there are uncertainties in modelling the indirect radiative forcing of the BC-containing particles, due to e.g. lack of knowledge about cloud interactions of these particles and the role of the co-emitted species. To overcome ...."

We have also added a brief discussion of the representativity of our BC particles to the revised manuscript (see our response to specific comment 3).

The BC particles were size selected with a DMA before being coated. This maintains the core constant while the coating thickness is increased. This approach is fine for the most part and produces interesting results. However, to untangle the effect of the coating hygroscopicity from that of size, it would have been interesting to size select before and after the coating stage as well, to maintain the overall particle size constant, while changing the coating thickness; in this way, isolating the effect of the overall particle size (this would be particularly interesting for the case of the oleic acid). I am not suggesting the authors should conduct such experiments for this manuscript, as I think the results of the current study are very interesting on their own, but they could briefly discuss this possibility for future studies.

Indeed, these kind of experiments would be an interesting topic for a future study. We have added a statement to the revised manuscript mentioning this possibility for future studies. However, they would be somewhat challenging (although perhaps not impossible) with the present setup, where the temperature of the furnace in effect determines the coating thickness. We selected the size of the BC cores first by a DMA and the coating thickness was varied by changing the furnace temperature. We then measured the size distribution of the coated particles exiting the furnace. At any given temperature, we then estimated the size of the coated particles using the peak value from the size distribution curves. Prompted by the comment by the other reviewer, we have also added a table specifying the temperatures used in the furnace in the revised manuscript.

**Specific comments**

5

20

25

Page 2 line 31, there are several studies, including some recent, that analyze and quantify the effect of coating, mixing and compaction for BC with the detail and unambiguity of electron microscopy, as well. Studies are available for both laboratory, as well as, ambient BC particles. It might be worth discussing some here.

We have modified the last paragraph of page 2 in the revised manuscript by changing the sentence in question according to our response to general comment 1.

- Page 3, line7, "of uncoated" seems not to belong here considering they are talking about multilayer models.

The reviewer is correct. We changed uncoated to pure particles in the first sentence of the paragraph. The adsorption activation model described by Sorjamaa & Laaksonen (2007) assumes that the CCN activation of insoluble but wettable compounds happens through multilayer adsorption of water molecules. This model was developed later by Kumar et al. (2011) to include the CCN activation of the insoluble particles coated by soluble salts.

- Page 3, line 27. Please provide a sentence or two on why regal back and how does that represent (how well it acts as) a surrogate for atmospheric BC. This in addition to the compaction issue mentioned in the general comments section.

We have added this clarification to the first paragraph of part 2.1. (page 4) of the revised manuscript:

"Regal Black is a surrogate for collapsed soot (Sedlacek et al. 2015) and is the recommended calibration standard for the SP-AMS (Onasch et al., 2012). This compound has been used in different studies (Onasch et al. 2012; Corbin et al. 2014; Healy et al. 2015; Sedlacek et al. 2015) as a model of refractory

- 15 carbonaceous compounds to estimate the chemical and physical properties of the black carbon particles. Canagaratna et al. (2015) have shown that regal black and flame soot appear very similar, at least from the perspective of the mass spectrometry. However, it should of course be borne in mind that in the ambient BC particles can vary significantly in terms of their physical and chemical properties, and is usually mixed with other pollutants present in the atmosphere.''
- Figure 5: the theoretical calculation seems to perform less well for the larger core diameter; in fact, the range of the grey band does not seem to intersect with the experimental data even considering their uncertainties. Is there any reason for that? A short discussion would be interesting.

This is a very good question, to which we do not have a definite answer. One reason might be just a larger uncertainty in the  $\kappa$  values than what is considered in the calculations. Different values have been reported for  $\kappa$  in different studies (Petters and Kreidenweis 2007; Chan et al. 2008; Petters et al. 2016). The  $\kappa$  we used for glutaric acid is from (Petters and Kreidenweis 2007) and is between 0.113-0.376 (we corrected it in the table 1).

We have also added the following sentence to the last paragraph of page 9 of the revised manuscript:

(')Nevertheless, there are small deviations between the measured and calculated critical supersaturations of the larger particles. The reason might be just a larger uncertainty in the  $\kappa$  values than what is considered in the calculations. For example, different values have been reported for  $\kappa$  of glutaric acid in different studies (Petters and Kreidenweis 2007; Chan et al. 2008; Petters et al. 2016).''

- Figure 6, maybe this was mentioned and I missed it, but why are the theoretical calculations so much narrower here than in figure 5? I guess the spread reflects directly the k range for the two coating materials, narrower for levoglucosan than for glutaric acid. Can the authors comment on that?

35

5

10

This is correct and we have thus added the following sentence to our revised manuscript, last paragraph on page 9:

"The uncertainty range of theoretical calculations for sc are narrower for levoglucosan compared for glutaric acid, because the variation in reported  $\kappa$  values used in the calculations, is smaller for levoglucosan (see Table 1)."

5

**Technical corrections**

- Page 3, line 13, consider adding the article "the" in front of Soot Particle Aerosol Mass Spectrometer

We have added "the" in front of Soot Particle Aerosol Mass Spectrometer.

10 - Page 10, line 7, "procudure", should be "procedure", probably.

We have corrected the word "procedure".

**Reviewer #2:** 15**

We thank Reviewer #2 for taking the time to consider and review our manuscript. We respectfully disagree, however, with the reviewer on the purpose and value of the presented work. We feel there is a fundamental difference of opinion in 1) what can be considered as "text-book knowledge"; 2) the value of well-defined laboratory studies with simple model systems for evaluating theoretical approaches used 20 to describe more complex atmospheric systems. Our point-by-point responses to the issues raised by the reviewer are below.

By reviewing the manuscript on the CCN activation properties of BC particles coated with a few organic species I feel myself rather uncomfortable. The study is a carefully planned and executed combination of experimental work and theoretical calculations aiming at providing new insights into the CCN 25 behavior of atmospheric BC particles. However, the entire approach looks as a textbook-like routine exercise that contains no traces of novelty and innovation that would have been required by a highstandard journal like Atmospheric Chemistry and Physics. Any laboratory study in itself is free to use virtually any combinations of agents and conditions, yet it would be expected to be a quasi-realistic model of physical reality. The basic concept of the present study does not fulfil this fundamental requirement.

30

We are sorry to hear that the reviewer feels uncomfortable, but thank him/her for acknowledging the quality of the presented work nevertheless. There seems to be a fundamental misunderstanding regarding the purpose and novelty of our work, however. As we write in the "Introduction", the CCN activation of uncoated and coated insoluble particles, such as BC-particles coated with soluble species, is usually

- 5 described theoretically by multilayer adsorption models accounting for the curvature of the particles. One of these theories is adsorption activation theory which is a combination of FHH adsorption isotherms and classical Köhler theory to describe the equilibrium growth of insoluble particles. Later, Kumar et al. (2011) introduced a new framework of CCN activation of dust containing a soluble salt fraction, based on a combination of the classical Köhler and FHH adsorption theories. However,
- 10 systematic experimental testing of the applicability of combined Köhler and FHH theory with agglomerated insoluble particles coated with organic species of varying solubility is lacking. Unlike what the reviewer claims, these theoretical approaches are not "well-established" in terms of systematically testing their applicability with known molecular species. This was the main motivation and novelty of our study and we have highlighted this now more clearly in the Introduction of the revised
- 15 *manuscript. We have added the following text to the paragraph 2 of page 3:*

*'However, systematic experimental testing of the applicability of combined Köhler and FHH theory with agglomerated insoluble particles coated with organic species of varying solubility is lacking.''*

We have also added the following sentence to the end of the first sentence at the last paragraph of page 3 of the revised manuscript:

" and to test the applicability of combined Köhler and FHH theory with agglomerated insoluble particles coated with organic species of varying solubility."

- 25 What sort of real-life BC particles do Regal black stand for? Aged diesel soot particles or BC particles from flaming biomass combustion? Levoglucosan, which is an abundant pyrolysis product of wood combustion, is not a semi-volatile species that is available for adsorption or condensation in the global atmosphere such as PAHs or n-alkanes. It is always present internally mixed with smoke particles, not as a gaseous species. Oleic acid is also a primary tracer which–unlike levoglucosan–is present in the gas phase but on a very limited spatial scale near its sources. I would doubt that this photochemically reactive species can make it to the free troposphere to participate in cloud nucleation. I wonder if anybody has ever detected oleic acid in cloud water or precipitation. In spite of these serious limitations oleic acid experiments at least yielded some unexpected results which are not fully exploited in the manuscript. Perhaps a molecular adsorption modelling approach would have helped explain the observations.
- 35 Glutaric acid does exist as SOA product in the atmosphere, though at far lower concentrations than smaller dicarboxylic acids or other SVOC species. In addition, a very similar study was published for the CCN effect of adipic acid. I suspect that upon releasing fresh BC particles from any source there is a plethora of co-emitted semi-volatile species that are ready to be adsorbed onto their surfaces. It is strange that in the experimental section no temperature values are given for the coating procedure. These
- 40 temperatures would also indicate that the atmospheric occurrence of such processes is unlikely.
  - 5

We are well-aware that in the atmosphere a myriad of different organic and inorganic compounds accompany BC in the particulate phase. The main purpose of this study was to test the model frameworks used to describe these extremely complex mixtures in various atmospheric models by comparing their predictions to well-defined particles generated in the laboratory. It is as well-established approach to

- 5 use laboratory measurements of model compounds representing the variation of properties relevant for the studied processes to evaluate theoretical frameworks. In fact, a major part of laboratory measurements represent this kind of approach. Hence we feel that the criticism raised by the reviewer is poorly justified and unfair.
- 10 As also pointed out in our responses to Reviewer #1, we added this clarification to the first paragraph of part 2.1. (page 4) of the revised

"Regal Black is a surrogate for collapsed soot (Sedlacek et al. 2015) and is the recommended calibration standard for the SP-AMS (Onasch et al., 2012). This compound has been used in different studies (Onasch et al. 2012; Corbin et al. 2014; Healy et al. 2015; Sedlacek et al. 2015) as a model of refractory carbonaceous compounds to estimate the chemical and physical properties of the black carbon particles.

- 15 carbonaceous compounds to estimate the chemical and physical properties of the black carbon particles. Canagaratna et al. (2015) have shown that regal black and flame soot appear very similar, at least from the perspective of the mass spectrometry. However, it should of course be borne in mind that in the ambient BC particles can vary significantly in terms of their physical and chemical properties, and is usually mixed with other pollutants present in the atmosphere.''
- 20 We would also like to highlight that the studied organic substances were chosen based on their properties, not solely based on their atmospheric relevance. The solubility and other properties of atmospheric organic material varies considerably (e.g. Goldstein and Galbally 2007; Jimenez et al. 2009) and this variation directed our choice.

We added the following text to the end of the first paragraph at part 2.1 (page 4)

25 ''It should be mentioned that the studied organic substances were chosen based on their properties, not solely based on their atmospheric relevance. The solubility and other properties of atmospheric organic material varies considerably (e.g. Goldstein and Galbally 2007; Jimenez et al. 2009) and this variation directed the selection of these material.''

We have also added Table 2 detailing the coating temperatures to the revised manuscript.

30 We have also added the following text to the last paragraph of page 11 in the revised manuscript:

"The oleic acid results simply highlight the need of laboratory measurements with simple model compounds. Our results showed that by using the existing model frameworks, we cannot explain all of the experimental observations, but more theoretical work is needed."

For lack of originality, the manuscript just declares plain trivialities such as on Page 9 Line 5-6 "As expected, the critical supersaturation is generally higher for pure BC particles than for the particles with

organic coating and the pure organic particles have the lowest critical supersaturation". Overall, this manuscript presents a lab-based approach in combination with a well-established theoretical approach that has little if any atmospheric relevance. It is a textbook-like repetition of previous studies and completely lacks originality and innovation.

- 5 First, we would like to point out that the fact that the present theories are shown to work well for two of the studied organic compounds does not mean that the results are not novel. Not all novel results need to be surprising or worrying in terms of the application of the theory for atmospherically relevant calculations. The reviewer claims that our study is a repetition of previous studies, but does not provide any references to back up these claims. We are not aware of any previous studies that investigate the
- 10 applicability of the adsorption-activation approaches for BC particles systematically coated with the studied organic compounds using a similar approach to the one presented here. We therefore find the criticism unjustified.

[revised manuscript text omitted]

---

## Author Response (AR2)

Dear Editor,

Thank you for considering our manuscript. We would also like to thank the reviewer for their contributions to improving it. Please find below our point-by-point responses to the reviewer comments along with the revised manuscript. The reviewer comments are written in normal font, and our responses in italics.

**Reviewer #1:**

*We would like to thank the reviewer for the careful consideration of our manuscript, and also for the encouraging comments regarding the novelty and importance of our study. We have made all the changes suggested by the reviewer, as itemized below.*

1. I think something is missing in the last sentence on the first page of the abstract (line 27 of page 10, specifically) starting with "Our results show potential ..." maybe I am just missing something, but I find it confusing.

*We have reformulated the sentence to "Our results show that present theories have potential for accurately predicting the CCN activity of atmospheric BC coated with organic species, given that the identities and amounts of the coating species are known."*

2. Line 29 of page 11 in the sentence starting with "Investigating ambient BC particles have shown..." I think "have" should be "has" if the verb refers to "investigating".

*Thank you for pointing this out. We have corrected this sentence.*

3. Line 16 of page 13, in the sentence starting with " However, it should of course be borne in mind that in the ambient BC..." maybe a comma should be put in front of "BC"? Or am I reading this incorrectly? Meaning, is "ambient" a noun here or is an adjective to BC? I think the authors mean it to be a noun, otherwise, I am missing the meaning of the sentence... Less important, but a comma might also be needed after "of course"

*Thank you for pointing this out. We have corrected this sentence.*

[revised manuscript text omitted]